# *MDGA1* Gene Variants and Risk for Restless Legs Syndrome

**DOI:** 10.3390/ijms26146702

**Published:** 2025-07-12

**Authors:** Félix Javier Jiménez-Jiménez, Sofía Ladera-Navarro, Hortensia Alonso-Navarro, Pedro Ayuso, Laura Turpín-Fenoll, Jorge Millán-Pascual, Ignacio Álvarez, Pau Pastor, Alba Cárcamo-Fonfría, Marisol Calleja, Santiago Navarro-Muñoz, Esteban García-Albea, Elena García-Martín, José A. G. Agúndez

**Affiliations:** 1Section of Neurology, Hospital Universitario del Sureste, 28500 Arganda del Rey, Madrid, Spain; hortalon@yahoo.es (H.A.-N.); alba.carcamo@salud.madrid.org (A.C.-F.); marisol.calleja@salud.madrid.org (M.C.); 2University Institute of Molecular Pathology Biomarkers, Universidad de Extremadura, 10003 Cáceres, Spain; soladeran@unex.es (S.L.-N.); payupar@unex.es (P.A.); elenag@unex.es (E.G.-M.); jagundez@unex.es (J.A.G.A.); 3Section of Neurology, Hospital La Mancha-Centro, 13600 Alcázar de San Juan, Ciudad Real, Spain; lturdoc@hotmail.com (L.T.-F.); jorge.millan.pascual@gmail.com (J.M.-P.); satinamu@hotmail.com (S.N.-M.); 4Movement Disorders Unit, Department of Neurology, Hospital Universitari Mutua de Terrassa, 08221 Terrassa, Barcelona, Spain; ignacioalvafer@gmail.com; 5Unit of Neurodegenerative Diseases, Department of Neurology, University Hospital Germans Trias i Pujol, The Germans Trias i Pujol Research Institute (IGTP), 08916 Badalona, Barcelona, Spain; pastorpau@gmail.com; 6Department of Medicine-Neurology, Universidad de Alcalá, 28801 Alcalá de Henares, Madrid, Spain; egarciaalbea@gmail.com

**Keywords:** restless legs syndrome, genetics, genetic polymorphisms, MDGA1 gene, risk factors

## Abstract

The MAM domain-containing glycosylphosphatidylinositol anchor 1 (*MDGA1*) gene, which encodes a protein involved in synaptic inhibition, has been identified as a potential risk gene for restless legs syndrome. A recent study in the Chinese population described increased MDGA1 methylation levels in patients with idiopathic RLS (iRLS) compared to healthy controls. In this study, we investigated the possible association between the most common variants in the *MDGA1* gene and the risk for iRLS in a Caucasian Spanish population. We assessed the frequencies of *MDGA1* rs10947690, *MDGA1* rs61151079, and *MDGA1* rs79792089 genotypes and allelic variants in 263 patients with idiopathic RLS and 280 healthy controls using a specific *TaqMan*-based qPCR assay. We also analyzed the possible influence of the genotype frequencies on several variables, including age at the onset of RLS, gender, a family history of RLS, and response to drugs commonly used in the treatment of RLS. The frequencies of the genotypes and allelic variants of the three common missense SNVs studied did not differ significantly between RLS patients and controls, neither in the whole series nor when analyzing each gender separately; were not correlated with age at onset and the severity of RLS assessed by the International Restless Legs Syndrome Study Group Rating Scale (IRLSSGRS); and were not related to a family history of RLS or the pharmacological response to dopamine agonists, clonazepam, or gabaergic drugs. Our findings suggest that common missense SNVs in the *MDGA1* gene are not associated with the risk of developing idiopathic RLS in Caucasian Spanish people.

## 1. Introduction

Restless legs syndrome (RLS) or Willis–Ekbom disease (WED), characterized mainly by sensory–motor symptoms, with well-established diagnostic criteria [1,2,3,4,5,6], is a highly prevalent neurological disorder [7,8,9,10]. The causative genes of RLS have not yet been fully identified. However, there is evidence suggesting an important role of genetic factors in its etiology. While 6 susceptibility genes were identified in initial genome-wide association studies (GWASs), a total of 21 susceptibility loci were identified in a further GWAS and meta-analysis, in addition to confirming the 6 previously described [11,12,13], and a recent review described up to 164 genetic risk loci for common and low-frequency variants [14]. However, these susceptibility sites would only explain approximately 11.3% of the heritability of RLS [12].

Iron deficiency and dopaminergic dysfunction seem to be the most important neurochemical features of RLS. Other neurotransmitter systems may also contribute (at least the glutamatergic, GABAergic, and adenosinergic systems), although they are not fully known [15].

The MAM domain-containing glycosylphosphatidylinositol anchor 1 (*MDGA1*) gene, located in chromosome 6p21.2 (gene ID 266727, MIM 609626), encodes a cell surface glycoprotein with the same name, which is predominantly expressed in the developing nervous system. This protein seems to play an important role in cell adhesion, migration, and axon guidance, and in the developing brain, it also plays an important role in neuronal migration (link https://www.ncbi.nlm.nih.gov.gene/266727, accessed on 9 July 2025). According to the data from the GTEx Portal (URL https://www.gtexportal.org/home/multiGeneQueryPage/MDGA1, last access on 5 July 2025), the *MDGA1* gene is predominantly expressed in the cerebellum and cerebellar hemisphere. It is also expressed in other neural tissues such as the hippocampus [16,17,18,19] and in human B-cells [20].

Several polymorphisms in the *MDGA1* gene are associated with the risk of schizophrenia [21,22] and bipolar disorder [22]. Moreover, the *MDGA1* gene has been found to be overexpressed in patients with major depressive disorder [23].

Based on the fact that in one genome-wide association study (GWAS), the *MDGA1* gene was shown to be one of the genes that comes with the potential risk of RLS [12] and considering the interaction of the *MDGA1* gene with the gene that showed the strongest association in GWASs for RLS (*MEIS1*) in some experimental models [24], Zhu et al. [25] conducted a study. They used two independent cohorts of patients with RLS and controls. They demonstrated increased levels of the methylation of the *MDGA1* gene in patients with iRLS compared to controls. They also found an association of the increased levels of the methylation of this gene with a positive family history for RLS. Their findings suggest an association between *MDGA1* gene methylation and the risk of developing RLS [25]. This suggests that the altered function of the *MDGA1* gene could be related to the risk of developing RLS.

In this study, we aimed to establish whether the most common missense single-nucleotide variants (SNVs) in the *MDGA1* gene of Caucasians were associated with the risk of RLS in Caucasian Spanish people.

## 2. Results

The genotype distributions for the three *MDGA1* single-nucleotide variants (SNVs)—rs10947690, rs61151079, and rs79792089—were in Hardy–Weinberg equilibrium in both the idiopathic restless legs syndrome (iRLS) patient group and the control group. Comparative analysis revealed no statistically significant differences in genotype or allelic frequencies between the 263 iRLS patients and 280 healthy controls (Table 1). This lack of association remained consistent when stratifying the data by sex (Appendix A).

For rs10947690, the most common genotype was A/A in both groups (67.7% in patients vs. 60.7% in controls), with no significant difference in the distribution of heterozygous (A/G) or homozygous variant (G/G) genotypes. Similarly, rs61151079 and rs79792089 showed no significant intergroup differences in genotype or allele frequencies. The minor allele frequencies for all three SNVs were low, particularly for rs79792089, where the A/A genotype was absent in both groups.

We further analyzed whether the presence of a positive family history of RLS influenced the distribution of *MDGA1* genotypes. Among the 259 patients with available family history data, 171 (65%) reported a positive family history. No significant differences in genotype or allele frequencies were observed between patients with and without a family history of RLS (Table 2). This suggests that the studied SNVs are not associated with a familial aggregation of the disorder.

To explore whether *MDGA1* variants influence the clinical phenotype of RLS, we compared the mean age at the onset of symptoms across different genotypes (Table 3). No statistically significant differences were found for any of the three SNVs. For example, the mean age at onset for rs10947690 A/A carriers was 42.54 years, compared to 46.01 years for A/G and 39.63 years for G/G carriers (*p* > 0.05 for all comparisons). Similar non-significant trends were observed for rs61151079 and rs79792089.

The severity of RLS symptoms, as measured by the International Restless Legs Syndrome Study Group Rating Scale (IRLSSGRS), did not significantly differ across genotypes for any of the three SNVs (Table 4). For instance, rs10947690 A/A carriers had a mean IRLSSGRS score of 24.06, compared to 25.19 for A/G and 24.75 for G/G carriers (*p* > 0.05). Although rs79792089 G/A carriers showed a numerically higher mean score (29.95), this difference did not reach statistical significance (*p* = 0.151).

We also assessed whether *MDGA1* genotypes influenced the therapeutic response to commonly used RLS treatments, including dopamine agonists, clonazepam, and GABAergic drugs. The response to these drugs was assessed both by the subjective improvement reported by the patients and the presence of a significant reduction (50%) in IRLSSGRS scores. No significant associations were found between genotype and treatment response (Appendix A), indicating that these SNVs do not appear to modulate pharmacogenetic outcomes in iRLS.

## 3. Discussion

The previous descriptions of the possible association between the *MDGA1* gene with the potential risk of RLS in GWASs [12], and the finding of increased methylation levels in this gene in patients diagnosed with iRLS, especially in those with a positive family history of RLS [25], make it reasonable to investigate the possible association between SNVs in this gene and the risk of RLS.

The results of the current study, which involved Caucasian Spanish people, did not show any associations of the three most common SNVs in the *MDGA1* gene (rs10947690, rs61151079, and rs79792089). In addition, none of these three SNVs were related to sex, the age of onset, or the severity of RLS, not even with the response of RLS symptoms to the most commonly used treatments for this condition.

The current study has several strengths, including a well-characterized cohort of iRLS patients diagnosed using standardized criteria and the use of robust genotyping methods. However, it also has limitations. The main limitation of the current study is that the sample size of the two analyzed cohorts is relatively small (for both iRLS patients and controls). Although this sample size should be appropriate for the detection of ORs of 1.5, it may not be sufficient to detect more modest associations. Taking into account this limitation, in this study, we failed to find any association between the three most common missense SNVs in the *MDGA1* gene and RLS risk in Caucasian Spanish people. The main results of this study, which are “negative”, fulfill the proposed standards of validity for studies with negative results, i.e., reporting primary outcomes, statistical power, and confidence intervals and show a plausible hypothesis [26,27]. The possibility that other SNVs in the *MDGA1* gene could be associated with the modification of the risk of RLS, as the alternative hypothesis, is not precluded by our results. Our findings suggest that future investigations should prioritize genome-wide or epigenome-wide approaches, possibly integrating methylation profiling, transcriptomics, and functional assays to better understand the role of *MDGA1* in RLS.

In conclusion, our results indicate that the three most common missense SNVs in the *MDGA1* gene are not associated with the risk of developing idiopathic RLS in Caucasian Spanish individuals. These findings underscore the complexity of RLS genetics and highlight the need for broader multi-omics studies to uncover the underlying biological mechanisms of this disorder.

## 4. Material and Methods

### 4.1. Patients and Controls

The current study involved 263 patients diagnosed with idiopathic RLS (iRLS) according to the International Restless Legs Syndrome Study Group (IRLSSG) diagnostic criteria [1], and 280 age- and sex-matched healthy controls were involved in this study. Approximately 60% of the patients included in the current study participated in several case–control genetic association studies previously reported by our group [11,28,29,30,31,32]. The exclusion of patients with diverse causes of secondary RLS, as was described in more detail elsewhere, was an obligate requisite for the diagnosis of iRLS [28]. Patients with iRLS were recruited from the Movement Disorders Units of the hospitals involved in this study, while healthy controls were recruited from students or staff of the University of Extremadura, and a mandatory requirement for inclusion in this study was the absence of a personal or family history of RLS and other movement disorders. Table 5 summarizes the clinical and demographic data from iRLS patients and controls.

### 4.2. Ethical Aspects

This study was approved by the Ethics Committees of the Hospital La Mancha-Centro (Alcázar de San Juan, Ciudad Real, Spain, 2016, no referral number), University Hospital “Príncipe de Asturias” (LIB 02/2017; Alcalá de Henares, Madrid, Spain), and the University Hospital of Badajoz (Badajoz, Spain, 2016, no referral number) and was conducted according to the principles of the Declaration of Helsinki.

### 4.3. Genotyping of MDGA1 rs10947690, MDGA1 rs61151079, and MDGA1 rs79792089 Variants

Genotyping studies were performed in genomic DNA obtained from the peripheral leukocytes of the venous blood samples of patients diagnosed with iRLS and controls. An analysis was performed by using real-time PCR (Applied Biosystems 7500 qPCR thermocycler, Foster City, CA, USA) with specific TaqMan probes (Life Technologies, Alcobendas, Madrid, Spain). The SNVs included in this study were selected according to their functional effect and allele frequencies in Caucasians (missense SNVs with minor allele frequencies higher than 0.01 in the population studied, according to the Genome aggregation database gnomAD) and were the following: (a) rs10947690 A/G (nonsynonymous, Leu61Pro, TaqMan assay id. C___3278725_10), (b) rs61151079 C/CACGAGG (nonsynonymous, Cys947_Ala948insProArg. Custom TaqMan assay id., and rs79792089 G/A (nonsynonymous, Ala942Val. TaqMan assay id. C___3278725_10). Apart from the three SNVs analyzed, other missense variants in the MDGA1 gene—such as rs75289615 (Gly926Glu) and rs192113659 (Glu718Asp)—exhibit extremely low minor allele frequencies (below 0.0002) in the population studied. Given the rarity of these variants, the likelihood of detecting them in either patients or controls was minimal. Moreover, even if identified, the statistical power would have been insufficient to draw meaningful conclusions. Therefore, we limited our analysis to the three most common SNVs to ensure the adequate power and reliability of the results. Appendix A summarizes the results of a cross-population comparison of SNV frequencies using data from gnomAD, indicating that common missense SNVs are more frequent in individuals of European ancestry. This higher frequency may facilitate the detection of associations with the risk of developing RLS in this population compared to other ethnic groups.

### 4.4. Statistical Analysis

SPSS version 27.0 for Windows (SPSS Inc., Chicago, IL, USA) was used to perform the statistical analysis. The Hardy–Weinberg equilibrium test was conducted with the online program https://www.snpstats.net/start.htm (last access on 31 May 2025), both in RLS patients and controls. Intergroup comparison values were calculated with the chi-square test or Fisher’s exact test where appropriate. We also calculated 95% confidence intervals, negative predictive values [33], and the correction for multiple comparison adjustments using the False Discovery Rate (FDR) [34].

We calculated the sample size using a genetic model to analyze the frequency of the lower allele with an odds ratio (OR) value = 1.5 (α = 0.05) from the allelic frequencies found in healthy subjects. The statistical power (two-tailed association) for variant alleles, according to the sample size of this study, was 82.44%% for rs10947690, 69.05% for rs61151079, and 10.34% for rs79792089. The SNV rs61151079 reached statistical power to detect an OR value of 1.7 (81.72%), whereas the rare SNV rs79792089 reached statistical power to detect an OR value equal to 3.9 (81.34%). These data are summarized in Appendix A.

Finally, the comparisons of the mean age at the onset of RLS symptoms and the severity of RLS symptoms according to the IRLSSG scale [35] between the different genotypes were performed by using a T-test for independent samples.

## Figures and Tables

**Table 1 ijms-26-06702-t001:** The genotypes and allelic variants of *MDGA1* gene in patients with RLS and healthy volunteers. The values in each cell represent numbers (percentage; 95% confidence intervals). P: crude probability; Pc: probability after multiple comparisons; NPV: negative predictive value.

GENOTYPE	RLS PATIENTS (N = 263, 526 Alleles)	CONTROLS (N = 280, 560 Alleles)	OR (95% CI), P; Pc; NPV (95% CI)
rs10947690 A/A	178 (67.7; 62.0–73.3)	170 (60.7; 55.0–66.4)	1.36 (0.95–1.93); 0.091; 0.459; 0.56 (0.51–0.62)
rs10947690 A/G	77 (29.3; 23.8–34.8)	99 (35.4; 29.8–41.0)	0.76 (0.53–1.09); 0.131, 0.459, 0.49 (0.46–0.52)
rs10947690 G/G	8 (3.0; 1.0–5.1)	11 (3.9; 1.7–6.2)	0.77 (0.30–1.94); 0.574, 0.670; 0.51 (0.51–0.52)
rs61151079 C/C	224 (85.2; 80.9–89.5)	229 (81.8; 77.3–86.3)	1.28 (0.81–2.02); 0.289; 0.640; 0.57 (0.47–0.66)
rs61151079 C/CACGAGG	37 (14.1; 9.9–18.3)	47 (16.8; 12.4–21.2)	0.81 (0.51–1.30); 0.382; 0.640; 0.51 (0.49–0.53)
rs61151079 CACGAGG/CACGAGG	2 (0.8; –0.3–1.8)	4 (1.4; 0.0–2.8)	0.53 (0.10–2.91); 0.457; 0.640; 0.51 (0.51–0.52)
rs79792089 G/G	259 (98.5; 97.0–100.0)	275 (98.2; 96.7–99.8)	1.18 (0.31–4.43); 0.809; 0.809; 0.56 (0.23–0.85)
rs79792089 G/A	4 (1.5; 0.0–3.0)	5 (1.8; 0.2–3.3)	0.85 (0.23–3.20); 0.809; 0.809; 0.52 (0.51–0.52)
rs79792089 A/A	0 (0.0; 0.0–0.0)	0 (0.0; 0.0–0.0)	--
**ALLELES**			
rs10947690 A	433 (82.3; 79.1–85.6)	439 (78.4; 75.0–81.8)	1.28 (0.95–1.73); 0.104; 0.360; 0.57 (0.50–0.63)
rs10947690 G	93 (17.7; 14.4–20.9)	121 (21.6; 18.2–25.0)	0.78 (0.58–1.05); 0.104; 0.360; 0.50 (0.49–0.52)
rs61151079 C	485 (92.2; 89.9–94.5)	505 (90.2; 87.7–92.6)	1.29 (0.84–1.97); 0.240; 0.360; 0.57 (0.47–0.67)
rs61151079 CACGAGG	41 (7.8; 5.5–10.1)	55 (9.8; 7.4–12.3)	0.78 (0.51–1.19); 0.240; 0.360; 0.51 (0.50–0.52)
rs79792089 G	522 (99.2; 98.5–100.0)	555 (99.1; 98.3–99.9)	1.18 (0.31–4.40); 0.810; 0.810; 0.56 (0.23–0.85)
rs79792089 A	4 (0.8; 0.0–1.5)	5 (0.9; 0.1–1.7)	0.85 (0.23–3.19); 0.810; 0.810; 0.52 (0.51–0.52)

**Table 2 ijms-26-06702-t002:** The genotypes and allelic variants of *MDGA1* gene in patients with RLS distributed by family history. The values in each cell represent numbers (percentage; 95% confidence intervals). P: crude probability; Pc: probability after multiple comparisons; NPV: negative predictive value.

GENOTYPE	POSITIVE FAMILY HISTORY OF RLS (N = 171, 342 ALLELES)	NEGATIVE FAMILY HISTORY OF RLS (N = 88, 176 ALLELES)	INTERGROUP COMPARISON VALUES *OR (95%CI) P, PC*
rs10947690 A/A	118 (69.0; 62.1–75.9)	57 (64.8; 54.8–74.8)	1.21 (0.70–20.9); 0.491; 0.696; 0.37 (0.28–0.46)
rs10947690 A/G	47 (27.5; 20.8–34.2)	30 (34.1; 24.2–44.0)	0.73 (0.42–1.28); 0.272; 0.696; 0.32 (0.28–0.36)
rs10947690 G/G	6 (3.5; 0.8–6.3)	1 (1.1; –1.1–3.4)	3.16 (0.38–26.70); 0.266; 0.696; 0.35 (0.33–0.35)
rs61151079 C/C	147 (86.0; 80.8–91.2)	73 (83.0; 75.1–90.8)	1.26 (0.62–2.54); 0.522; 0.696; 0.39 (0.25–0.54)
rs61151079 C/CACGAGG	22 (12.9; 7.8–17.9)	15 (17.0; 9.2–24.9)	0.72 (0.35–1.47); 0.363; 0.696; 0.33 (0.30–0.35)
rs61151079 CACGAGG/CACGAGG	2 (1.2; 0.4–2.8)	0 (0.0; 0.0–0.0)	1.52 * (0.30–1.52); 0.309; 0.696; 0.34 (0.34–0.34)
rs79792089 G/G	168 (98.2; 96.3–100.2)	87 (98.9; 96.6–101.1)	0.64 (0.06–6.28); 0.703; 0.703; 0.25 (0.01–0.78)
rs79792089 G/A	3 (1.8; 0.2–3.7)	1 (1.1; –1.1–3.4)	1.55 (0.16–15.16); 0.703; 0.703; 0.34 (0.33–0.35)
rs79792089 A/A	0 (0.0; 0.0–0.0)	0 (0.0; 0.0–0.0)	--
**ALLELES**			
rs10947690 A	283 (82.7; 78.7–86.8)	144 (81.8; 76.1–87.5)	1.07 (0.66–1.71); 0.792; 0.812; 0.35 (0.26–0.45)
rs10947690 G	59 (17.3; 13.2–21.3)	32 (18.2; 12.5–23.9)	0.94 (0.58–1.51); 0.792; 0.812; 0.34 (0.32–0.36)
rs61151079 C	316 (92.4; 89.6–95.2)	161 (91.5; 87.4–95.6)	1.13 (0.58–2.20); 0.714; 0.812; 0.37 (0.23–0.52)
rs61151079 CACGAGG	26 (7.6; 4.8–10.4)	15 (8.5; 4.4–12.6)	0.88 (0.46–1.71); 0.714; 0.812; 0.34 (0.32–0.35)
rs79792089 G	339 (99.1; 98.1–100.1)	175 (99.4; 98.3–100.5)	0.76 (0.08–7.36); 0.812; 0.812; 0.25 (0.01–0.78)
rs79792089 A	3 (0.9; –0.1–1.9)	1 (0.6; –0.5–1.7)	1.55 (0.16–15.00); 0.704; 0.812; 0.34 (0.34–0.34)

* The relative risk is shown instead of the odds ratio because one group in the comparison has a value equal to 0.

**Table 3 ijms-26-06702-t003:** Age at onset of RLS according to genotypes.

	Age at Onset (SD) Years	Two-Tailed T-Test Compared to A/A	Two-Tailed T-Test Compared to A/G
rs10947690 A/A	42.54 (17.55)		
rs10947690 A/G	46.01 (16.33)	0.141	
rs10947690 G/G	39.63 (13.80)	0.644	0.290
		**Two-Tailed T-Test Compared to C/C**	**Two-Tailed T-Test Compared to C/CACGAGG**
rs61151079 C/C	44.28 (17.04)		
rs61151079 C/CACGAGG	39.06 (19.42)	0.096	
rs61151079 CACGAGG/CACGAGG	32.00 (29.70)	0.314	0.626
		**Two-Tailed T-Test Compared to G/G**	**Two-Tailed T-Test Compared to G/A**
rs79792089 G/G	43.34 (17.04)		
rs79792089 G/A	45.50 (20.44)	0.803	
rs79792089 A/A	--	--	--

**Table 4 ijms-26-06702-t004:** IRLSSGRS scores of RLS patients according to genotypes.

	IRLSSG (SD)	Two-Tailed T-Test Compared to A/A	Two-Tailed T-Test Compared to A/G
rs10947690 A/A	24.06 (6.52)		
rs10947690 A/G	25.19 (7.30)	0.223	
rs10947690 G/G	24.75 (4.37)	0.769	0.867
		**Two-Tailed T-Test Compared to C/C**	**Two-Tailed T-Test Compared to C/CACGAGG**
rs61151079 C/C	24.24 (6.63)		
rs61151079 C/CACGAGG	25.17 (7.00)	0.444	
rs61151079 CACGAGG/CACGAGG	27.50 (6.36)	0.489	0.649
		**Two-Tailed T-Test Compared to G/G**	**Two-Tailed T-Test Compared to G/A**
rs79792089 G/G	24.39 (6.71)		
rs79792089 G/A	29.95 (4.57)	0.151	
rs79792089 A/A	--	--	--

**Table 5 ijms-26-06702-t005:** Demographic and clinical data of series studied.

Group	RLS (n = 263)	Controls (n = 280)
**Age (years), mean (SD)**	56.0 (14.6)	55.4 (15.7)
**Age at onset (years), mean (SD)**	43.8 (17.3)	NA
**Female %**	204 (77.6%)	217 (77.5%)
**Positive family history %**	171 (65.0%) ^1^	NA
**IRLSSG, mean (SD)**	24.74 (6.39)	NA

^1^ Family history was recorded for 259 out of the 263 patients.

## Data Availability

All data relating to the current study, intended for reasonable use, is available from J.A.G. Agúndez (University Institute of Molecular Pathology Biomarkers, University of Extremadura -UNEx ARADyAL Instituto de Salud Carlos III, Av/de la Universidad S/N, E10071 Cáceres. Spain) and F.J. Jiménez-Jiménez (Section of Neurology, Hospital del Sureste, Arganda del Rey, Madrid, Spain).

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
