# Peer review of "MDGA1 Gene Variants and Risk for Restless Legs Syndrome"

_ijms, 2025, doi:10.3390/ijms26146702_

Round 1
Reviewer 1 Report
Comments and Suggestions for Authors
This article investigates the association between MDGA1 gene variation and restless legs syndrome (RLS), contributing valuable insights to both academic research and clinical practice. The following comments are provided for the author’s reference, with the aim of further enhancing the scholarly quality and impact of the paper.
Major Issues
The study included 263 patients with idiopathic RLS (iRLS) and 280 control subjects. Given the relatively modest sample size—particularly in the context of genetic association studies—the ability to detect associations with moderate or weak effect sizes (e.g., OR < 1.5) may be limited. Although this limitation is acknowledged by the authors, a formal power analysis was not conducted to quantify the actual detection capacity of the current sample. It is therefore recommended that the authors perform a power calculation to determine the minimum detectable OR under the given sample size and significance level, and discuss how the risk of false-negative findings might affect the interpretation of the results.
While the article highlights the potential link between MDGA1 methylation identified by Zhu et al. and RLS, the present study focuses solely on single nucleotide variants (SNVs) without incorporating epigenetic or functional data such as DNA methylation levels, gene expression, or protein abundance. To strengthen the mechanistic interpretation, it is suggested that targeted methylation analysis be performed or that publicly available datasets (e.g., GTEx) be referenced to explore the regulatory patterns of MDGA1 in relevant neural tissues.
The study reports analyses of three SNVs but does not specify whether corrections for multiple comparisons were applied (e.g., Bonferroni or FDR methods). If no correction was implemented, the statistical threshold should be adjusted accordingly (e.g., α = 0.017 instead of 0.05), and the possibility of false-positive results should be explicitly addressed in the discussion.
Minor Issues
The article cites a methylation study by Zhu et al. conducted in a Chinese population but does not consider whether the observed negative findings could be influenced by race-specific genetic backgrounds. A cross-population comparison of SNP frequencies (e.g., using data from the 1000 Genomes Project) would help assess genetic heterogeneity across ethnic groups and improve the generalizability of the conclusions.
There is inconsistency in the gene nomenclature throughout the manuscript; "MDAG1" appears once (line 25 of the abstract), while "MDGA1" is used elsewhere. It is recommended that the correct official symbol "MDGA1" be consistently adopted to avoid confusion.
The results section lacks specific statistical values (e.g., p-values, ORs, 95% CIs), relying instead on qualitative descriptions such as "no significant difference." To enhance transparency and reproducibility, it is advised to include a table or supplementary materials detailing genotype distributions, allele frequencies, and corresponding statistical test outcomes for each SNP.
Other Issues
The abstract states that "No association was found between SNV and drug responses such as dopamine agonists," yet the main text does not describe the analytical methods employed, including criteria for evaluating treatment response or adjustments for covariates. Providing detailed methodological information and supporting data (e.g., response rates, effect sizes) will strengthen the validity of this conclusion.
Email formats listed in the author affiliations exhibit inconsistencies—for example, "jorge.millan.pascual@gmail.com" contains an underscore, whereas others do not. A standardized format should be adopted for all email addresses.
Several section headings appear to have incorrect numbering (e.g., "3. Discussion 185" is immediately followed by "4. Patients and Methods"). The overall structural coherence of the manuscript should be reviewed and corrected accordingly.
The introduction contains several lengthy sentences (e.g., lines 44–53), which may hinder readability. It is recommended that these be divided into shorter, more concise statements to improve clarity and comprehension.
Author Response
COMMENTS ARE INCLUDED IN A PDF FILE

Reviewer 2 Report
Comments and Suggestions for Authors
Restless legs syndrome (RLS) is one kind of prevalent neurological disorders featured by sensory-motor symptoms. Even though this disease has well established diagnostic standards, the causative genes of risk for RLS remains largely unknown. Recently, a report studied by Zhu et al. suggested increased level of methylation of the MDGA1 gene in patient with iRLS is associated with the risk of developing RLS. Inspired by this new study, this manuscript investigated 3 most missense single nucleotide variants to check their relations to RLS genesis. Their results indicated that these common missenses in the gene MDGA1 are not with the causal relationship of developing RLS. Despite all results presented negative, this manuscript may still make progress of understanding this disease. However, I have some comments that need to be addressed before it can be accepted for publication.
Major comments
- In this manuscript, authors just studied the 3 single nucleotide variants in the gene MDGA1. Why authors choose such variants rather than other types of variants? Because so many different variants are existent, will more gene-related variants help identify potential risk of developing RLS?
- Although these results are negative forms showing no significant statistics, it will contribute to this field if authors can compare these results to extract phenotype changes between patients and healthy control. There is no figure form in this manuscript, so it will be helpful to show figure results considering the convenience.
Minor comments
- Line 85, Table 2 should be Table 1? Or should include Table 1?
- Manuscript number line overlapped with these Tabel contents.
- In Table 1, what is y?
Author Response
COMMENTS ARE UPLOADED IN A PDF FILE

Round 2
Reviewer 1 Report
Comments and Suggestions for Authors
Accept.
Reviewer 2 Report
Comments and Suggestions for Authors
Concerns were addressed, and I recommend acceptance for publication.